



Lidar-based modelling approaches for estimating solar insolation in heavily forested streams
Richardson, Jeffrey J.*[1]; Torgersen, Christian E.[2]; and Moskal, L. Monika[3]
[1] *Sterling College, Craftsbury Common, VT, USA*
[2] *U.S. Geological Survey, Forest and Rangeland Ecosystem Science Center, Cascadia Field Station,*
*University of Washington, Seattle, WA, USA*
[3] *Precision Forestry Cooperative, School of Environmental and Forest Science, University of Washington,*
*Seattle, WA, USA*
*Corresponding Author
Abstract
Methods to quantify solar insolation in riparian landscapes are needed due to the importance of stream
temperature to aquatic biota. We have tested two approaches developed for other applications of
estimating solar insolation from airborne lidar using field data collected in a heavily forested narrow
stream in western Oregon, USA.  We show that a raster methodology based on the light penetration
index (LPI) and a synthetic hemispherical photograph approach both accurately predict solar insolation,



explaining more than 73% of the variability observed in pyranometers placed in the stream channel. We
apply the LPI based model to predict solar insolation for an entire riparian system, and demonstrate that
no field-based calibration is necessary to produce unbiased prediction of solar insolation using airborne
lidar alone.
A.    Introduction
Accurately quantifying solar insolation, defined as the amount of solar radiation incident on a specific
point on the Earth's surface for a given period of time, is essential to a diversity of ecological
applications. In forested ecosystems, trees interact with solar radiation through shading, and thus solar
insolation at fine spatial scales in these systems can vary widely. Understanding the heterogeneous
patterns of insolation below tree canopies has been important for numerous applications, such as
understanding the importance of sunflecks for understory photosynthesis, gaining insight into the
patterns of seedling regeneration in dense forests (Nicotra et al., 1999), and explaining patterns of
snowmelt (Hock, 2003) and soil moisture (Breshears et al., 1997).
The relationship between stream temperature and solar insolation is of particular interest in this study,
as high amounts of solar energy intercepting a stream can cause adverse ecological effects, which can in
turn limit options for forest management near streams. In northwestern North America, a large amount
of research has focused on the relationship between forest practices, stream temperature, and the
corresponding effect on river salmonid fishes (Holtby, 1988;Leinenbach et al., 2013;Moore et al.,
2005a;Moore et al., 2005b).  Direct measurement of stream temperature with in-stream thermographs
can be used to quantify thermal diversity (Torgersen et al., 2012;Torgersen et al., 2007), but ground-
based measurements are time consuming, expensive, and impractical for large areas. In addition, stream
temperature measurements can only show the effect of forest management practices if taken before





and after trees are removed. In order to predict the potential effect of forest management practices on
stream temperature, models may be needed to estimate the amount of solar insolation intercepting
streams using remotely sensed data (Forney et al. 2013).
Several different methods have been utilized for measuring or predicting solar insolation on the ground.
Pyranometers are the most direct method for measuring insolation, capturing the solar radiation flux
density above a hemisphere as an electrical signal and cataloguing those signals in a datalogger (Kerr et
al., 1967). Once calibrated, these signals give a measure of the total direct and diffuse solar radiation
intercepting a point for a given period of time (Bode et al., 2014;Forney et al., 2013;Musselman et al.,
2015). While pyranometers give direct measurement of solar insolation for a defined period of time,
hemispherical photographs allow indirect estimation of solar insolation for any point in time (Bode et
al., 2014;Breshears et al., 1997;Rich et al., 1994). Plotting the path of the sun in the area of sky captured
by the hemispherical photograph allows for calculation of direct solar radiation through identified
canopy gaps, while gap fraction across the entire hemisphere allows for calculation of diffuse radiation.
Analysis of hemispherical photographs requires assumptions of solar output and sky conditions in order
to produce solar insolation estimates. Understory light conditions can also be modeled by creating a
three-dimensional reconstruction of a forest from field-based biophysical measurements (Ameztegui et
al., 2012) or terrestrial laser scanning (Ni-Meister et al., 2008). All ground-based measurements are
limited by the time and cost required to collect data, and thus solar insolation can only be calculated for
relatively small spatial extents.
Airborne and satellite remote sensing methods provide a means for estimating solar insolation over
large spatial extents. Satellite-based methods utilizing passive remote sensing data can provide coarse-
scale estimates of solar radiation absorbed by tree canopies through radiative transfer models based on
spectral indices  (Field et al., 1995;Asrar et al., 1992), but these methods are not suitable for fine-scale



application such as modeling stream temperature. Airborne lidar is the preferred method for
characterizing three-dimensional structure of forest canopies, and thus is also used to assess the
shading effect of those canopies. Below we discuss three different approaches that have been used in
previous studies to quantify solar insolation at ground level using aerial lidar.
*Raster Approaches*
Lidar data can be used to create raster datasets by selecting various attributes of lidar points within a
defined spatial neighborhood around a raster cell. One of the most common raster products for
assessing canopy structure is the light penetration index (LPI), the ratio of ground first return points
(typically less than 2 m in elevation above ground) to the total number of lidar first return points within
a given raster cell. This ratio has been shown to be useful for characterizing light extinction in canopies
according to the Beer-Lambert law (Richardson et al., 2009) and thus has been explored as a predictor of
understory light conditions (Musselman et al., 2013;Alexander et al., 2013;Bode et al., 2014). GIS
software solar radiation calculators can also be used to compute solar insolation on a lidar-derived
digital elevation model (DEM). Bode et al. (2014) combined a GRASS r.sun solar insolation estimation
based on a DEM with LPI to produce estimates of ground level solar insolation that showed high
accuracy compared to pyranometer-collected field data in a mixed forest in Northern California, USA.
*Lidar Point Reprojection*
Lidar point returns can be reprojected from the X,Y,Z Cartesian coordinate system in which they are
most often delivered by a vendor into a spherical coordinate system which centers the point cloud
around a specific location on the ground. This reprojection allows for a circular graph of the lidar point
returns to be created around a point at ground level. Alexander et al. (2013) created a canopy closure
metric from these projected point graphs based on gap fraction, and found that this metric was
correlated to Ellenburg indicator values of understory light availability. Moeser et al. (2014) created



synthetic hemispherical photographs from reprojected lidar returns, and solar irradiance at ground level
was calculated using traditional hemispherical photograph analysis software.  The processed synthetic
hemispherical photographs showed good correlation to pyranometer measured solar irradiance at three
field sites in eastern Switzerland.
*Point Cloud Approaches*
Because lidar point clouds are typically represented in a three-dimensional Cartesian coordinate system,
it is possible to model the sun's position in relation to that three-dimensional space.  The number of
lidar returns that are reflected from a defined volume between the direction of the sun and the ground
can then be calculated. These methods are computationally intensive, but have shown promise for
providing the most direct measure of understory light availability. Lee et al. (2008) calculated the
number of points within a conical field of view directed at the sun's location and created a model to
relate this to ceptometer measurements of photosynthetically active understory solar radiation at
specific times and locations in a pine forest in northern Florida, USA.  This method is limited by its
reliance on raw lidar point counts specific to the actual and relative point densities within their lidar
acquisition. Raw point counts are affected by both changes in flight characteristics between missions,
and the patterns of flight line overlap within a mission. A different point cloud approach involves a linear
tracing of the sun's rays along their path to the ground, and Martens et al. (2000) demonstrated how a
ray-tracing algorithm could be used to characterize understory light conditions in a computer simulated
forest. Peng et al. (2014) combined a lidar-based ray tracing algorithm with field-collected canopy base
heights to produce an estimate of understory solar insolation based on the Beer-Lambert law that
compared well to field-collected pyranometer data but is limited in practical application because of its
reliance on field- measured data in its model.  Musselman et al. (2013) used a ray-tracing algorithm to
produce highly detailed estimates of direct beam solar transmittance in 5-minute increments by



voxelizing the lidar data and summing the number of voxels that a ray intercepted between the point of
origin and the sun. The algorithm relied on site specific pyranometer measurements to calibrate and
adjust the beam transmittance, and therefore we were restricted from testing this method in this study.
Our objectives were to test the accuracy and precision of established methods of quantifying solar
insolation from aerial lidar within areas of narrow, heavily forested streams. We utilized the raster
approach and the lidar point reprojection approach, two methodologies that had not been previously
applied and tested using high quality field data collected in heavily forested streams. We evaluated the
two methods by comparing  model results to field-based pyranometer measurements of solar insolation
and hemispherical photograph-based measures of shade in Western Oregon, USA. Further, we sought to
apply this method to quantify solar insolation throughout a small headwater stream network.

B.   Methods
*Study Site*
All field locations were located within the wetted channel of Panther Creek and a tributary (Figure 1) in
narrow streams (1-6 m in width) located in the east side of the Coast Range of Oregon, USA within a
larger research area in which lidar has been used to quantify forest canopy structure (Flewelling and
McFadden, 2011). All field sites were within a mature Douglas-fir (*Pseudotsuga menziesii*) forest, with
other dominant trees including red alder (*Alnus rubra*), Western red-cedar (*Thuja plicata*), and Western
hemlock (*Tsuga heterophylla*). The elevation profile and description of the stream can be found in
(Richardson and Moskal, 2014). The center of the channel was manually digitized as a polyline in ArcGIS
using a combination of aerial imagery and the vendor-provided lidar DEM.



Four transects were installed in late June 2015 using a Leica Builder Total Station and georeferenced
using a Javad Maxor GPS unit. The locations of the transects can be seen in Figure 1, with the 19 point
locations used for capturing field data denoted by black dots surrounded by white circles (A contains 3
points, B and C contain 4 points, and D contains 8 points). Transect locations were chosen manually in
order to maximize variability in forest shade while allowing for safe access by the field crew.  Each point
location was located within the stream channel and marked by driving rebar into the substrate until only
1 m was exposed above the water surface. Point locations were approximately 15 m apart within a
transect in order to allow data from multiple point locations to be collected by a single datalogger.
Two datasets were collected at each point location. A hemispherical photograph was collected using a
Nikon CoolPix 4500 digital camera leveled on a tripod 1 m above the ground under uniform sky
condition (Figure 2) utilizing a method to find the optimum light exposure (Zhang et al., 2005). Each
hemispherical photograph was analyzed using the Gap Light Analyzer (GLA) program (Frazer et al., 1999)
in order to produce estimates of percent transmittance for diffuse and direct sunlight. An Apogee
Instruments SP-110 self-powered pyranometer, leveled and mounted to the rebar pole at 1 m height
(Figure 3) was used to collect a full day's solar output at each point location using the datalogger. The
raw voltage values collected by the datalogger were calibrated to solar irradiance using the closest
publicly available meteorological data. All pyranometer datasets were collected on cloudless days,
except for transect A, and pyranometer data from transect A was not used in this study. The calibrated
pyranometer data from a point location from transect D is shown in Figure 4.




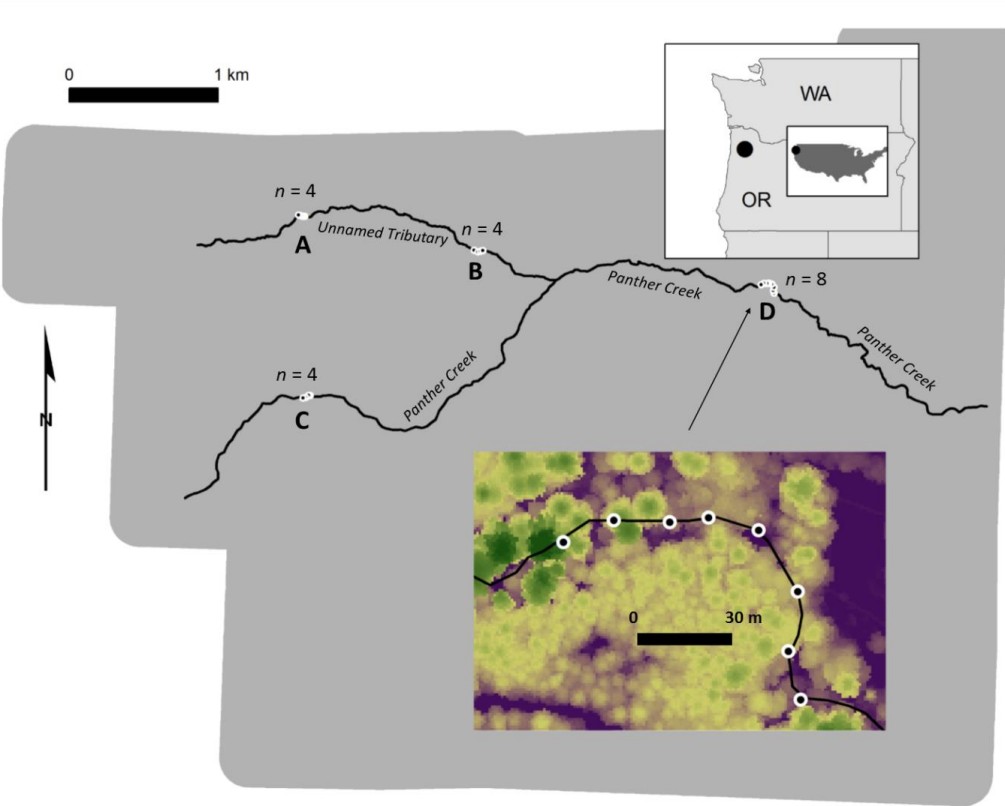



*Figure 1: Study area in northwestern Oregon (USA). The grey polygon is the extent of the 2015 lidar*
*acquisition. The black circles surrounded by white circles represent the 19 point locations. The letters A,*
*B, C, and D denote the four transects. The inset shows transect D and the background raster in the inset*
*is the lidar derived canopy height model with green representing tall trees and purple representing the*
*lowest heights. The direction of flow is from west to east.*





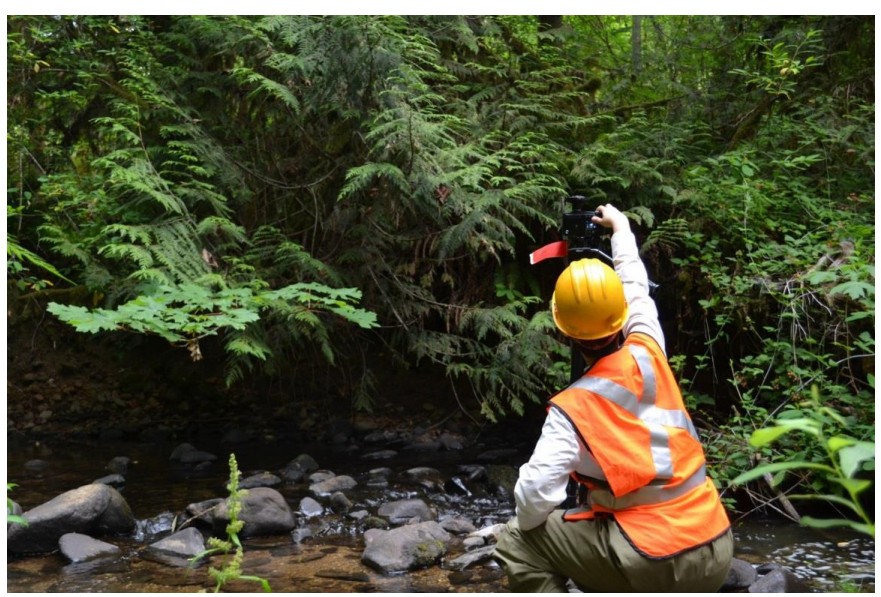


*Figure 2: Example of hemispherical photograph acquisition at a plot location in transect D.*

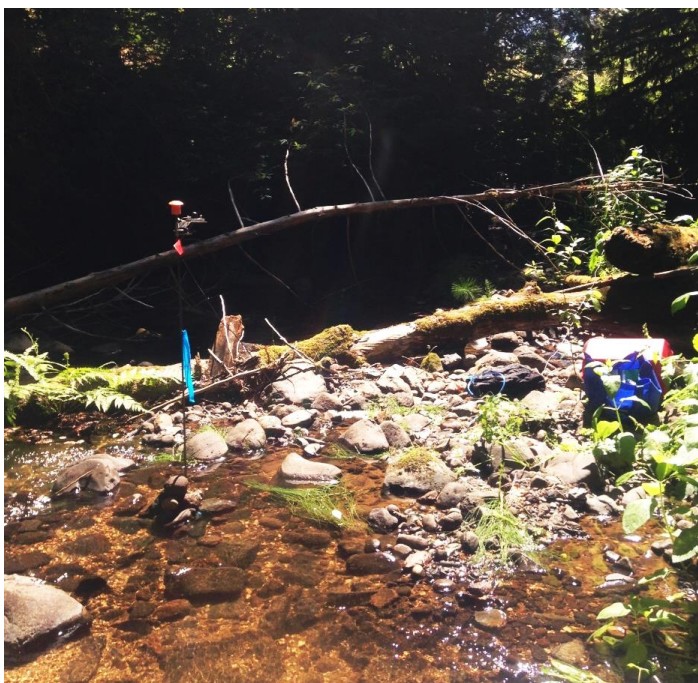


*Figure 3: Example of pyranometer installation at transect D (note that pyranometer is mounted on south*
*side of pole at a height of 1 m).*





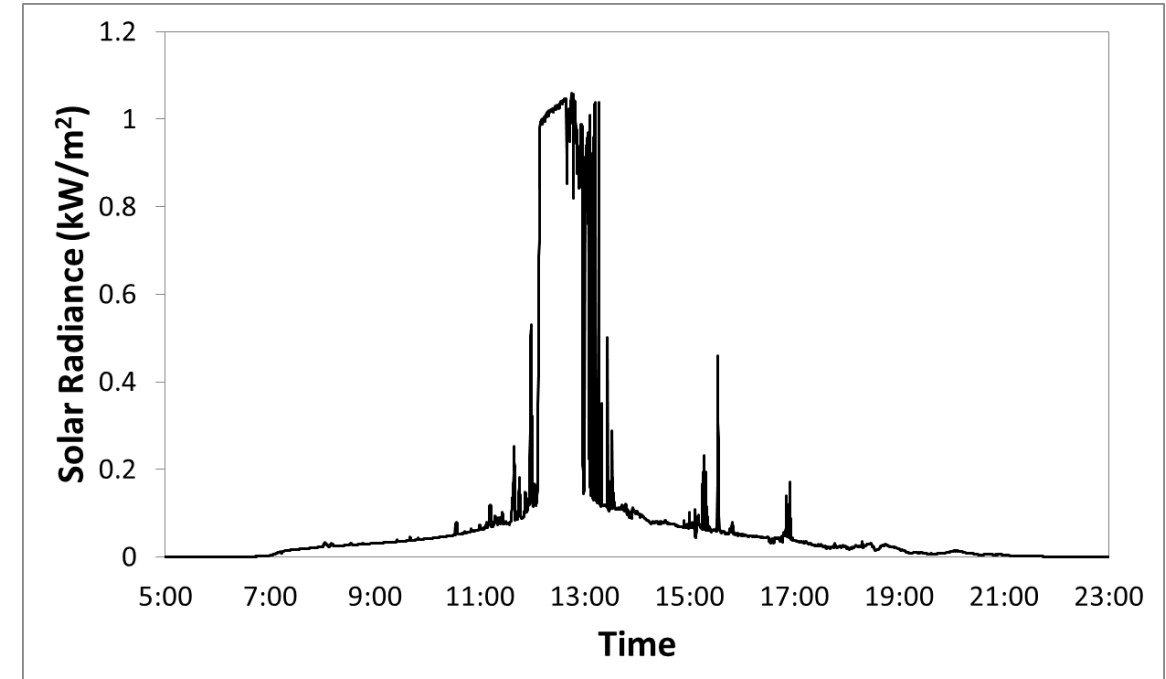


*Figure 4: Daily pyranometer output from sunset to sundown for a plot in Transect D*

*Lidar Data and Analysis*
Airborne discrete-return lidar was acquired in June of 2015 according to the specifications described in
Table 1. The vendor provided processed discrete lidar point returns as well as a lidar DEM and highest
hit model at a pixel resolution of 1 m. The highest hit model was subtracted from the DEM to create a
canopy height model (CHM) describing the vegetation height normalized to the ground surface. In
addition, FUSION (McGaughey, 2009)  was used to subtract the elevations of the raw lidar points from
the ground elevation in the DEM to produce a normalized point cloud dataset (NPCD).

*Table 1: Lidar Data Specifications*





| Acquisition Date | June 18, 2015 |
|---|---|
| Sensor | Leica ALS80-HP |
| Survey Altitude | 1,400 m |
| Pulse Mode | MPiA (Multiple Pulses in Air) |
| Pulse Rate | 394.8 kHz |
| Field of View | 30 degrees |
| Mean Pulse Density | 25.35 pulses/m$^2$ |
| Overlap | 100% with 65% sidelap |
| Relative Accuracy | 4 cm |
| Vertical Accuracy | 5 cm |



Effective Leaf Area Index ($L_e$) was computed using the NPCD according to the method in Richardson et
al. (2009) :
$L_e = -\frac{1}{k} \ln(R_g/R_t)$
Where $k$ is the extinction coefficient equal to 2, $R_g$ is the number of first ground returns and $R_t$ is the
number of total first returns. LPI was computed as:
$LPI = (R_g/R_t)$
$L_e$ and $LPI$ were computed in ArcGIS using a circular buffer with radius 10 m around each field point
location mirroring the radius used in Richardson et al. (2009) . $LPI$ was also computed using a shifted
square buffer modified from the method of Bode et al. (2014) where the buffer side length ($s$) was
calculated based on:



$$s = \frac{h}{\tan \theta}$$

Where $h$ is equal to the modal tree height across all our plots (34 m), and $\vartheta$ is equal to the maximum
lidar scan angle subtracted from 90° (75°), resulting in a buffer side length of 9.12 m. The square buffer
was shifted south to account for the seasonal solar angle in the northern hemisphere according to:

$$shift = \left(\frac{s}{1 + \cos \sigma}\right) - s$$

Where $\sigma$ is equivalent to the solar angle at noon on the date of interest. A solar angle of 68° was used
in this study, resulting in a southern shift of 3.42 m. We also computed topographically influenced solar
radiation using the lidar DEM and the solar radiation function in ArcGIS, but found that there was no
significant difference across the plot locations and thus did not use these results in subsequent analysis.
Synthetic hemiphotos were created in Matlab using the method of Moeser et al. (2014) and analyzed for
diffuse and direct light transmittance in GLA. All statistical analyses were performed in R (version 3.4).
Longitudinal profiles of stream shading were created in ArcGIS in 1-m increments based on the
intersections of the stream polyline centerline with the raster output of modeled solar insolation.
C.   Results and Discussion
*Comparison between Pyranometers and Hemispherical Photographs*
Figure 5 shows the correlation between field-collected pyranometer data and processed hemispherical
photographs, with data from transect A removed.  These data are highly correlated ($r^2 = 0.87$), but these
data are also not equally distributed across a range of solar insolation. Many more plot locations were at
low levels of solar insolation than in areas of relatively low shade. This is very typical of the heavily



forested streams in northwestern North America. Note that none of our plot locations contained
transmittance greater than 40%.

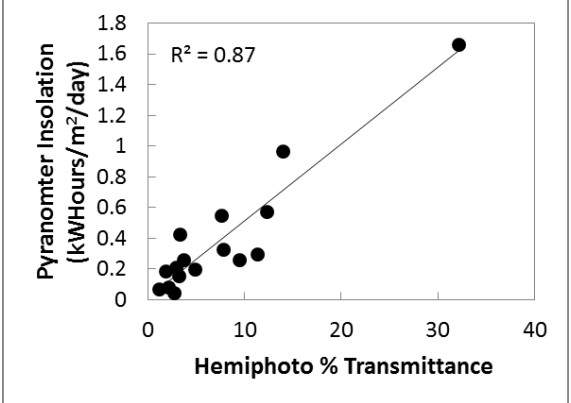


*Figure 5: Comparison between pyranometer-measured solar insolation and daily diffuse and direct radiation*
*canopy transmittance calculated from hemispherical photographs.*

*Model Comparisons*

Pyranometer-based solar insolation and hemispherical photograph percent diffuse and direct radiation
transmittance calculated at all point locations except transect A were compared to a variety of
predictors using simple linear regression. These results are shown in Figure 6. Effective LAI was not
highly correlated to either predictor, showing a non-linear relationship. The LPI calculated using a 10 m
circle centered on the point location explained about 55% of the variability in both response variables,
but the prediction accuracy significantly improved when LPI was calculated using the shifted square
buffer. Shifted LPI explained 74% of the variability in solar insolation and 64% of the variability in
percent transmittance. Synthetic hemispherical photographs explained 77% of the variability in solar





insolation and 60% of the variability in percent transmittance. Figure 6 shows comparisons between
transects B, C, and D to make interpretation easier, but Table 2 shows the results of linear regressions
between predicted variables and hemispherical photograph transmittance for all plot locations resulting
in small reductions in the amount of variability explained. Table 3 gives model parameters of slope and
intercept resulting from the simple linear regression.




*Figure 6: Simple linear regressions between predictor variables and field measured pyranometer solar insolation*

233        *(A, C, E, G) and hemispherical photograph % transmittance (B, D, F, H) omitting data from transect A*







*Table 2: Coefficients of determination for the simple linear regression between predictor variables and*
*hemispherical photograph transmittance using three additional point locations from transect A*

| Predictor Variable | Coefficient of Determination ($r^2$) |
|---|---|
| Effective Leaf Area Index | 0.32 |
| Light Penetration Index | 0.54 |
| Shifted Light Penetration Index | 0.54 |
| Synthetic Hemispherical Photograph % Transmittance | 0.45 |















Table 3: Model parameters from simple linear regressions. Note that all regressions are significant (p < 0.05). Data
from transect A are excluded.

| Response Variable | Predictor Variable | Slope | Intercept |
|---|---|---|---|
| Hemispherical Photograph % Transmittance | Effective Leaf Area Index | -3.40 | 25.26 |
| | Light Penetration Index | 124.09 | -3.29 |
| | Shifted Light Penetration Index | 142.2 | -4.49 |
| | Synthetic Hemispherical photograph % Transmittance | 1.01 | -0.32 |
| Pyranometer Insolation | Effective Leaf Area Index | -0.19 | 1.37 |
| | Light Penetration Index | 6.73 | -0.19 |
| | Shifted Light Penetration Index | 8.23 | -0.30 |
| | Synthetic Hemispherical Photograph % Transmittance | 0.07 | -0.08 |



While both the raster-based shifted LPI approach and the lidar point reprojection synthetic
hemispherical photograph approach achieved satisfactory model performance, the limited range of
solar insolation conditions at the point locations in our study limits some of the conclusions that can be
drawn. Excluding transect A, 14 of the 16 point locations received less than 0.8 kWHours/m$^2$/day,



leading to the other two point locations to exert a large degree of leverage on the model results. The
three points in transect A all received less than 0.8 kWHours/m$^2$/day and their inclusion in the model
results (Table 2) did not improve model results, suggesting that all models are not as effective at
predicting field measured values in areas of high canopy cover. The constraints of the study design
requiring point locations to be located in the stream made it impossible to achieve a greater range in
solar insolation. It is reasonable to expect that including more point locations receiving larger amounts
of insolation would have led to improved model accuracy and greater coefficients of determination, as
previous studies have shown that accuracy increases as canopy cover decreases (Moeser et al.,
2014;Musselman et al., 2013;Richardson and Moskal, 2014). In areas with no canopy and thus no lidar
point returns above the ground, the models should show better agreement with field measurements.

One explanation of the decrease in variability explained by the models at high canopy cover is
demonstrated in Figure 7. Here, a synthetic hemispherical photograph from transect D is compared to a
field-captured hemispherical photograph with the GLA modeled sunpath superimposed. This sunpath is
critical for determining the quantity of direct light, but very small differences in the center location of
the two images can produce large differences in the modeled direct light. The sunpath passes through a
modeled canopy gap near solar noon on the synthetic hemispherical photograph, while it intersects only
canopy and misses the gap on the field-collected hemispherical photograph. Very small registration
errors can cause significant differences in transmittance at low light levels, and we suggest that these
errors are likely to cause the errors observed in the models.

Understory vegetation is another likely cause of observed errors, as airborne lidar is inherently limited in
its ability to fully sample multi-layered canopies (Richardson and Moskal, 2011). We noticed several
points with significant differences to the model results that contained understory vegetation in close



proximity to the field instruments. The ideal scenario would be for the lidar scan angles to precisely
match the range of potential solar angles at each plot location, but this is currently impractical, leading
to an incomplete sample of the canopy light environment which contributes to the errors observed in
the models.

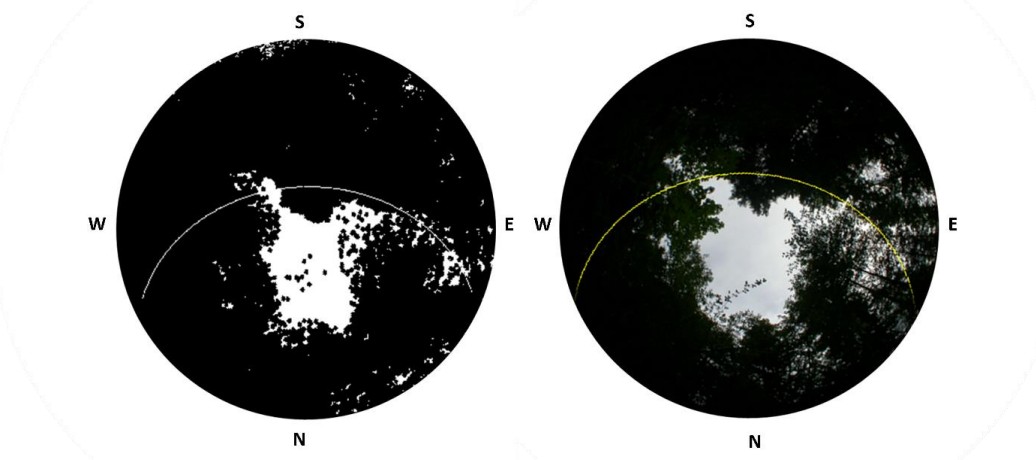


*Figure 7: Sunpath superimposed on a synthetic hemispherical photograph (left) and a field acquired hemispherical*
*photograph (right) at a point location in Transect D. The letters represent the four cardinal directions.*

*Model Application*

Model G and Model E (Figure 6) performed the best and are both appropriate to use as the basis for
estimating solar insolation across the study area. Implementation of Model G was the simplest and least
time-intensive method, and we chose to modify Model G by multiplying LPI by the maximum above
canopy solar insolation for June 20, 2015 and then computing a non-intercept linear regression (Figure
8). Removing the intercept from the model lowered the coefficient of determination but provided a
model with very little bias, only slightly underestimating model insolation. Figure 9 shows the model
applied across the study area. The graphs show the pattern of solar insolation across the two reaches in




the study, highlighting the utility of these methods for predicting solar insolation in heavily forested
streams across wide spatial extents. Figure 10 shows the relative frequency of binned solar insolation
values, highlighting the dominance of heavily shaded areas (note that a dammed reservoir, point D on
the map, contributes the majority of the points in full sun).

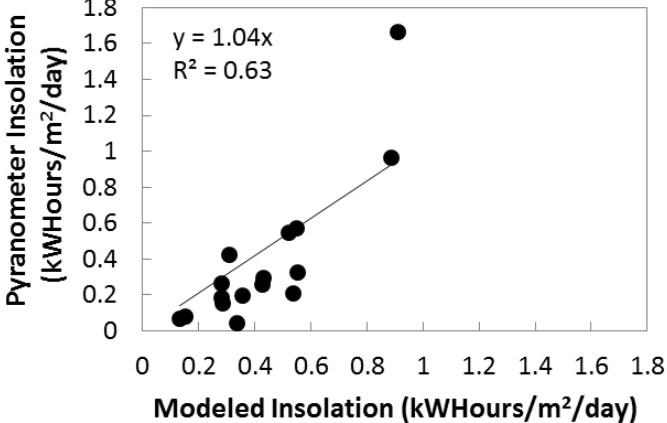


307                    *Figure 8:* Model used for generation of landscape scale solar insolation estimates








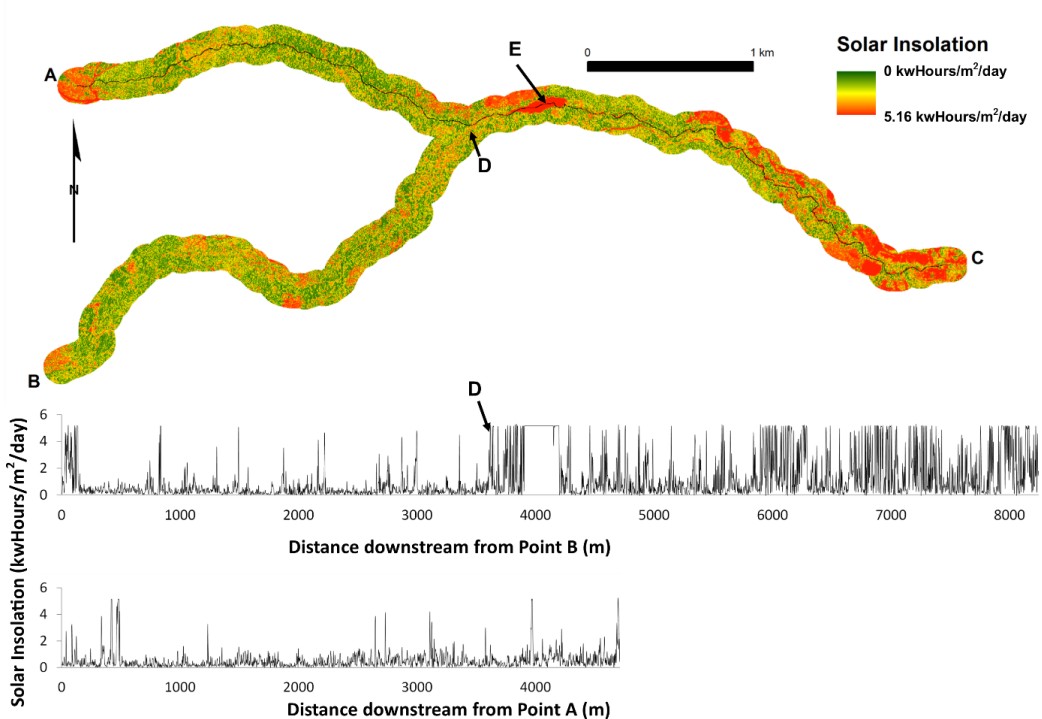



*Figure 9: Map of model derived solar insolation for Panther Creek (top) and graph of model derived solar*
*insolation for reach A-C (middle) and reach B-D (bottom). Point E is a dammed reservoir. Note the*
*direction of flow is toward point C*

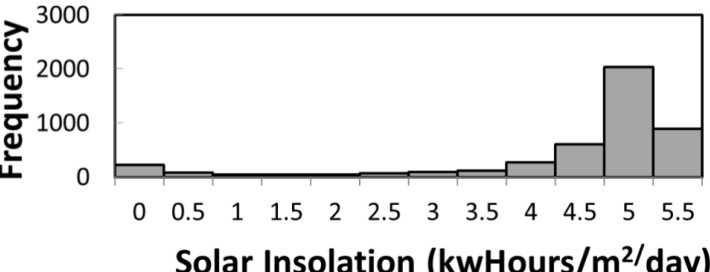


*Figure 10: Histogram of solar insolation pixel values along reach A-C from Figure 9*






The relatively unbiased results shown in Figure 8 show that field calibration is not required to produce
accurate estimates of solar insolation. However, information is still needed on local above-canopy
meteorological conditions, which can either be modeled from known solar outputs or collected from a
nearby meteorological station.  Little bias was observed in comparisons between synthetic
hemispherical photograph transmittance and field-based hemispherical photograph transmittance
(Table 3). Therefore, both approaches tested in this study should not require field calibration.

D.  Conclusions
We tested two approaches for estimating solar insolation from airborne lidar using field data collected
in a heavily forested narrow stream, showing that an LPI-based raster approach and a synthetic
hemispherical photograph approach accurately predict solar insolation and light transmittance. These
results should be interpreted with the caveat that our point locations contained few areas with high
insolation. We showed that the LPI-based model can be applied across the landscape, and we
demonstrated that no field-based calibration was necessary to produce unbiased prediction of solar
insolation.
This study lays the groundwork for additional research on remote sensing methods for quantifying light
conditions in riparian areas over heavily forested streams. First, point-cloud based approaches utilizing
ray-tracing need to be further developed. The results of this study suggest that refined ray-tracing
approaches should not require calibration. Ray-tracing is perhaps the most elegant method for
accurately modeling the relationship between lidar points and the sun, but this method requires a large
amount of computational power to model multiple sun angles for each lidar point. Second, research
should focus on exploring the limit of matching ground-based measurements to lidar-predicted solar

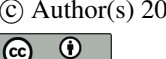



insolation. Lastly, the limitation of aerial lidar to quantify understory light conditions in multi-layered
canopies should be explored in more detail to better understand when and if airborne sensors are
inappropriate for these particular applications. In these circumstances, other sensors such as terrestrial
lidar or ground-based digital photographs utilizing structure from motion may provide additional useful
information.
E.  Data availability
The GPS data, pyranometer data, processed hemispherical photograph data, spreadsheets used for data
analysis, and access to the LiDAR data can be found at https://doi.org/10.17632/vwmxw4hcj7.1
F.  Acknowledgements
We are grateful to Dave Moeser for sharing his MATLAB code for creating synthetic hemispherical
photographs and to Keith Musselman for advising on the applicability of ray-tracing methods. Guang
Zheng also assisted with research into ray-tracing methods. Caileigh Shoot and Natalie Gray coordinated
field data collection. This work was supported by the Precision Forestry Cooperative, the Bureau of Land
Management, and the U.S. Geological Survey. Any use of trade, product or firm names is for descriptive
purposes only and does not imply endorsement by the U.S. government.

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
