# Peer review of "Lidar-based approaches for estimating solar insolation in heavily forested streams"

_Hydrology and Earth System Sciences, 2018_

## Referee Comment (RC1) · Collin Bode (Referee) · 21 Mar 2019

GENERAL COMMENTS Full disclosure: I am the author of one of the models evaluated in this paper, e.g. the raster 'shifted LPI' model (Bode et al., 2014).

The paper compares two models subcanopy light models to validation datasets. The author's interest is in applying the modeled light to stream temperature studies in heavily forest areas.

Evaluation papers like this are a critical part of assessing the utility of environmental models. The use of external validation datasets is an excellent way to do this. So, the

general approach of this paper has scientific merit and a solid methodology.

The introduction is excellent, clearly laying out the case for the value of quantifying subcanopy insolation and in reviewing current literature on modeling efforts.

On three substantive issues I have concerns:

Model vs predictor. The abstract clearly states this paper is testing two models with two validation datasets. However, under Model Comparisons, the discussion changes to four "predictors" without explanation how these relate to the two models or why effective leaf area index is included, as it is part of neither model. This confusion is compounded under Model Application, where the predictors are now referred to as Model G and Model E, in reference to graphs in figure 6. More consistent naming from methods through the discussion would make this easier to follow.

Pyranometer validation. The spectral response of silicon-cell photodiodes is calibrated to clear sky direct sunlight conditions, because it is not sensitive to the full shortwave spectrum and responds to various wavelengths with different intensities. Leaf shading selectively blocks certain wavelengths, which causes silicon pyranometers to decalibrate. Apogee estimates that this produces roughly a 19% error under conifer canopy (https://www.apogeeinstruments.com/content/SP-100-200-spec-sheet.pdf, page 15). Black body thermopile pyranometers are recommended for subcanopy light measurements. They have an even spectral response across the shortwave spectrum even under leaves. I recommend the authors acknowledge this as a source of uncertainty in their discussion.

Conclusions. Line 256 "While both the raster-based LPI approach and the lidar point reprojection synthetic hemispherical photograph approach achieve satisfactory model performance, the limited range of solar insolation conditions at the point locations in our study limits some of the conclusions that an be drawn." While I appreciate this study and the intent behind it, perhaps more validation data is needed? Was there insufficient information to effectively evaluate the two models? How are both approaches

satisfactory?

SPECIFIC COMMENTS

Line 146: The dates are not given for when the pyranometers were recorded. This makes a significant difference for the models. On June 20, summer solstice, the shifted LPI and general LPI will look almost identical, but December 20, winter solstice, will look radically different. Is there a reason this is not mentioned, while the date for the Lidar is mentioned?

Line 251: Table 3 linear regression slope and intercept. I think this can be removed without loss to the paper.

Line 269: Models should agree better in areas without shading. I am not sure how this is a conclusion. While true, the whole point of these models is to tackle the uncertainty of heavily shaded landscapes.

Line 271: small registration errors. Recommend identifying which model this is an error for. Relevant for synthetic photo, but not for raster.

Line 281: understory vegetation. This is actually an argument against the directions this paper recommends on Line 335 regarding ray tracing. Note the raster approach was developed with this issue as one of the problems it was solving in its design.

Line 294: "Model G and Model E (figure 6) performed the best..." This statement is unclear. How are plots models? What criteria states that they performed the best? Their performance and the performance of the hemispherical photos all seem within error of each other. Is this incorrect?

Line 337: "The results of this study suggest that refined ray-tracing approaches should not require calibration." I do not see this statement supported by the paper. Both models used in this study did not perform point cloud ray tracing. That is their strength. Musselman and Lee (referenced in introduction) used voxel ray-tracing. Both required calibration.

---

## Referee Comment (RC2) · Anonymous Referee #2 · 21 Mar 2019

The study presents an interesting approach to derive solar insolation estimates at and near forested riverbanks from Lidar data. Other than point-based measurements with pyranometers or hemispherical photographs, this allows for a spatially continuous mapping taking topography and vegetation geometry into account. The authors compare two Lidar-based estimates to measured references at 19 points in four transects along the Panther Creek NW Oregon, USA. They conclude both methods to be feasible for heavily forested stands, although areas with high insolation due to more open vegetation have not been covered by the study.

While the study is generally well-written and will likely become an interesting contribution to HESS, I find some of its structure and arguments to require and deserve revisiting. In the following I will outline my major concerns and suggestions. I hope the authors will find these as constructive as I intend them to be.

**1 General comments:**

I appreciate very much that the authors provide their data and analysis (as is HESS standard now). While I could easily follow the general setup of the study, I found it difficult to grasp the information residing in the Lidar data set and how it has been used. Since the latter is not included in the repository: Did I understand correctly that the Lidar data was commercially acquired and preprocessed to 1m pixels? So each pixel has values about all point returns, the number of highest hits (canopy) and the number of lowest hits (ground)?

Please be more specific about the calculation methods than naming the Software ArcGIS. I suppose this is an array operation which could be done in R (or any other math software) too. Which approaches did you employ? What can be understood about the "10m Buffer around the field points" (L187) and how does it differ to the "shifted square buffer" (L188f.)? Did you average within this area for comparison? What are the effects on the performance of the estimates. Especially with regards to the issue of "registration errors" L277ff. would this mean that a higher resolution could be more accurate or in other words that the hemispherical photographs suffer from minor shading effects to become representative at stand scale?

For a validation of the Lidar-derived solar insolation there is basically the correlation plot in Fig. 8 comparing it to pyranometer measurements. To me this does not appear very convincing to support the conclusion. By not allowing for an intercept

in your linear regression model, you define the bias-term to be zero. While this is an understandable desire in comparing two measurements which should give the same results, I do not understand your statement in L298f. The 16 points appear to overestimate the pyranometer references in most cases. High insolation references are underestimated. With an R2 of 0.63, I find it rather problematic to speak of accurate: L329f. "a synthetic hemispherical photograph approach accurately predict solar insolation and light transmittance".

In this respect, I moreover have difficulties to relate this back to the presented indices which leaves me with a couple of questions about the reason of their introduction in the first place. This confusion might partially stem from the mannifold usage of the term "model" in the manuscript. I would suggest to allow for a more precise terminology to differentiate regression analyses from conversion models, from indices and from spatial map models. From the title I was expecting several modelling approaches using the Lidar data, which I did not find in the manuscript.
Coming back to the indices (Fig. 6, Tab. 3) I do not find the focus of the study specifically suitable to address these correlations. Contrastingly, the comparison of synthetic and actual hemispherical photograph (Fig. 7) is very compelling but falls in my view a little short in its analysis and evaluation (e.g. applying this for all 16 locations).

Since the validation of the "Lidar-based modelling" is rather difficult using the 16 measurements alone, maybe some further reference could be derived from remote sensing products? This could also provide the link to some of the addressed indices?

**2   Minor comments:**

**L28f.:** why only ecological applications?

**L29:** do trees really interact (so having feedbacks) with solar radiation?

**L36:** can (solar) energy intercept with something? maybe irradiate a stream?

**L37:** how does solar irradiation limit options for forest management? I do not understand.

**L48ff.:** is it really necessary to describe the function of a pyranometer (at this broad level of detail)?

**L53:** I do not see the difference between the time references of a direct state measurement and the photograph

**L56:** Depending on the type of pyranometer, diffuse radiation is directly measured too.

**L67:** Start new paragraph with "Airborne lidar..." ?

**L113f.:** very confusing. please rephrase.

**Fig 1:** I would prefer all four Lidar models/maps instead of the grey box, which I assume to be the total Lidar dataset footprint. If you find my suggestion feasible, maybe a map of a satellite RS derived index could also be a reference here. A colourbar would be nice.

**L200f.:** What happened to the longitudinal profiles? Were they processed?

**L215:** See general comment. Which exactly are THE models? do you refer to the different indices? the calculus to derive them? a model to generate the synthetic hemispherical what are the assumptions behind the comparison approach? What is the observation reference deemed as closest to the true value?

**L257:** model performance? in reference to what? Is a R2 to each other really a good measure?

**L277ff.:** I do not understand why this should not be desirable... actually, i find the results in fig 7 quite convincing and the sensitivity might be quite an interesting feature. Pls. see my general comment on this, too.

---

## Referee Comment (RC3) · Anonymous Referee #3 · 21 Mar 2019

Summary

The authors present an interesting study that compares two LiDAR based techniques (i.e., a raster-based method and a synthetic hemispherical photograph approach) for estimating under canopy solar insolation, which is an important variable for predicting stream temperature dynamics. They conduct their study for sites on the heavily forested Panther Creek and its tributary located in Oregon, USA While I am generally supportive of the merits of the study the authors present, I believe they could be more precise in their language and provide more connecting details about the methods used so that their work can be replicated and advanced. I also have some specific con-

cerns about the methods in the models. Additionally, throughout the paper, there is an emphasis on the ecological implications of this work. However, stream temperature also has important implications for various biogeochemical processes. The work the authors present may be of interest to other research domains so I would recommend that the authors broaden their discussion to encompass them. I have provided some general comments and suggestions that I hope the authors will consider incorporating into their paper to address the problems I have enumerated.

General Comments

1. While the authors indicate that they used two LiDAR based approaches/models for estimating solar insolation, midway through the paper, they introduce the new term "predictors" and then switch back to models (Line 294). This is confusing. I would suggest that the authors select one term and consistently use it throughout the paper. I would actually recommend sticking to predictor since they are essentially correlating various shading surrogate indexes with measurements of solar insolation. I also think it will be good introduce the specific predictors used under each approach (i.e., raster & synthetic hemispheric photograph approaches) at the beginning of the paper so that their introduction later in the paper is not so abrupt. Under raster-based predictors they could introduce LPI, SLPI, and LAI and then introduce %Transmittance for hemispheric photograph approach. They could also discuss why they are good/suggested predictors for solar insolation citing references.

2. The authors conclude that the limitation of their study was the lack of more monitoring points with large insolation values and that inclusion of more of these points would have increased the model accuracy (Line 266), but the point of their study was to derive approaches for estimating solar insolation for streams with heavily forested riparian zones. This is in practice the areas where insolation estimation uncertainty is greatest. My recommendation is to make this their focus and perhaps remove the points with higher insolation values from their regression.

[Figure]

3. Throughout the paper, the authors use the word "significant" to describe differences between values conjuring up an image of statistical significance. I would recommend that the authors state the actual numerical differences or use other words.

4. While the connection between solar insolation is self-apparent. I would recommend making that connection more explicit in the paper. You could say something along the lines of "Solar radiation is a major source heat flux into streams providing up to y% of heat fluxes" and the then cite a reference.

5. For the synthetic hemispherical photographs, what resolution was used for the hemisphere? Did it match the field photographs? If different, what are the implications of the differences for the authors analysis. I think the comparison of these too and the reasons why they might differ is an important contribution.

Specific Comments

1. Line 16 – "due to the importance of temperature to aquatic biota". This makes it sound like aquatic biota is the only reason why quantifying solar insolation is important. Consider revising to broaden its implications.

2. Line 17-19: I suggest changing "two approaches..." to something like "four predictor indexes computed using two approaches for estimate shading effects from LiDAR" or something along these lines. The larger point is that it is important to be precise in describing what was actually done.

3. Line 28 "is essential to a diversity of ecological..." Again, I think you can broaden this.

4. Line 36 "solar energy intercepting a stream..." Consider revising to "solar energy irradiating a stream"

5. Line 36-37 "can in turn limit options for forest management". Could the authors explain how increasing temperatures limit options for forest management? I am not sure this is true.

[Figure]

6. Line 45-46 "models may be needed..." I would argue that this is actually often the approach that is used and is not a new insight so please consider revising to "models are therefore often employed to estimate temperature"

7. Line 57: "solar output" consider revising to extra-terrestrial solar radiation.

8. Line 60: "All ground-based..." Sounds a little too strong. Consider removing "All".

9. Line 78-79. "GIS software solar radiation calculators..." Consider revising to "Solar radiation calculators in GIS software"

10. Line 80-82. I think you are missing some words somewhere. Please rephrase for clarity. E.g., "r.sun solar insolation model for the GRASS GIS software..."

11. Line 89: What are Ellenburg indicator values? While ecologist might be familiar with them, I think it will be good to explain.

12. Line 169 Figure 4: Does the y axis name need to be solar irradiance for consistency?

13. Line 195-197: I am not sure why this sentence is part of the paper. I feel it is unnecessary. Please consider removing.

14. Line 198-199: Are the authors able to delve more into the details of the creation of these synthetic photos?

15. Line 222. "significantly improved" remove significantly for the reasons I raised earlier.

16. Line 278: Please remove the word "significant". for the same reasons as before.

17. Line 298-299: I am not sure I am comfortable removing the intercept and saying the resulting model has little bias. By removing the intercept, the authors are making the $R^2$ value no longer useful.

18. Line 311 & Figure 9: Please consider adding an inset that zooms to one of the

monitoring points.

19. Line 337-340: The authors pivots to ray tracing. However, the methods they use does not include any ray tracing.

---

## Short Comment (SC1) · 14 Apr 2019

On three substantive issues I have concerns: Model vs predictor. The abstract clearly states this paper is testing two models with two validation datasets. However, under Model Comparisons, the discussion changes to four "predictors" without explanation how these relate to the two models or why effective leaf area index is included, as it is part of neither model. This confusion is compounded under Model Application, where the predictors are now referred to as Model G and Model E, in reference to graphs in figure 6. More consistent naming from methods through the discussion would make this easier to follow.

[Figure]

AGREED THAT THIS IS CONFUSING AND IMPRECISE. THE FINAL VERSION WILL BE EDITED TO CLARIFY THE EXACT PREDICTORS USED IN THE ABSTRACT, METHODS, RESULTS AND DISCUSSION.

Pyranometer validation. The spectral response of silicon-cell photodiodes is calibrated to clear sky direct sunlight conditions, because it is not sensitive to the full shortwave spectrum and responds to various wavelengths with different intensities. Leaf shading selectively blocks certain wavelengths, which causes silicon pyranometers to decalibrate. Apogee estimates that this produces roughly a 19% error under conifer canopy (https://www.apogeeinstruments.com/content/SP-100-200-specsheet.pdf, page 15). Black body thermopile pyranometers are recommended for subcanopy light measurements. They have an even spectral response across the shortwave spectrum even under leaves. I recommend the authors acknowledge this as a source of uncertainty in their discussion.

THANK YOU FOR POINTING THIS OUT. WE WILL ADD THIS SOURCE OF UNCERTAINTY TO THE DISCUSSION.

Conclusions. Line 256 "While both the raster-based LPI approach and the lidar point reprojection synthetic hemispherical photograph approach achieve satisfactory model performance, the limited range of solar insolation conditions at the point locations in our study limits some of the conclusions that an be drawn." While I appreciate this study and the intent behind it, perhaps more validation data is needed? Was there insufficient information to effectively evaluate the two models? How are both approaches satisfactory

AGREED THAT "SATISFACTORY" IS NOT WELL-DEFINED AND THUS THIS STATEMENT IS NOT VERY USEFUL. WILL REWORD TO INDICATE THAT THE RESULTS MAY BE SATISFACTORY DEPENDING ON THE APPLICATION BUT MORE VALIDATION DATA IS NEEDED. C2

SPECIFIC COMMENTS Line 146: The dates are not given for when the pyranometers

were recorded. This makes a significant difference for the models. On June 20, summer solstice, the shifted LPI and general LPI will look almost identical, but December 20, winter solstice, will look radically different. Is there a reason this is not mentioned, while the date for the Lidar is mentioned?

THIS WAS AN OVERSIGHT. PYRANOMETER AND HEMIPHOTO DATA WERE COLLECTED OVER TWO WEEKS AROUND THE SUMMER SOLSTICE IN 2015. THIS INFORMATION WILL BE ADDED TO THE METHODS.

Line 251: Table 3 linear regression slope and intercept. I think this can be removed without loss to the paper.

THIS IS INCLUDED FOR COMPLETENESS SAKE AND BECAUSE CERTAIN SCATTER PLOTS IN FIGURE 6 (eg. G AND H MIGHT BE DIFFICULT TO INTERPRET WITHOUT THE INCLUSION OF A 1:1 LINE)

Line 269: Models should agree better in areas without shading. I am not sure how this is a conclusion. While true, the whole point of these models is to tackle the uncertainty of heavily shaded landscapes.

THAT SENTENCE WILL BE REMOVED

Line 271: small registration errors. Recommend identifying which model this is an error for. Relevant for synthetic photo, but not for raster.

AGREED. WILL INCLUDE IN REVISED VERSION

Line 281: understory vegetation. This is actually an argument against the directions this paper recommends on Line 335 regarding ray tracing. Note the raster approach was developed with this issue as one of the problems it was solving in its design.

GOOD OBSERVATION AND AGREED. WILL REMOVE RAY TRACING FROM THE CONCLUSION EXCEPT TO NOTE THAT FURTHER RESEARCH IS NEEDED.

Line 294: "Model G and Model E (figure 6) performed the best..." This statement is

unclear. How are plots models? What criteria states that they performed the best? Their performance and the performance of the hemispherical photos all seem within error of each other. Is this incorrect?

THIS CONCLUSION WAS BASED ON THE COEFFICIENT OF DETERMINATION. WE ARE CONSIDERING THE SIMPLE LINEAR REGRESSION AS THE MODEL.

Line 337: "The results of this study suggest that refined ray-tracing approaches should not require calibration." I do not see this statement supported by the paper. Both models used in this study did not perform point cloud ray tracing. That is their strength. Musselman and Lee (referenced in introduction) used voxel ray-tracing. Both required calibration.

AGREED. THIS SENTENCE WILL BE REMOVED AND RAY TRACING WILL BE RE-MOVED FROM CONCLUSION EXCEPT FOR SHORT STATEMENT ON FURTHER RESEARCH.

---

## Short Comment (SC2) · 14 Apr 2019

I appreciate very much that the authors provide their data and analysis (as is HESS standard now). While I could easily follow the general setup of the study, I found it difficult to grasp the information residing in the Lidar data set and how it has been used. Since the latter is not included in the repository: Did I understand correctly that the Lidar data was commercially acquired and preprocessed to 1m pixels? So each pixel has values about all point returns, the number of highest hits (canopy) and the number of lowest hits (ground)?

THE LIDAR WAS PRE-PROCESSED BY THE VENDOR INTO 1 M PIXELS CONTAIN-

ING HIGHEST ELEVATION AND GROUND MODEL. THE AUTHORS CREATED ADDITIONAL RASTERS USING THE RAW LIDAR POINTS IN ORDER TO DETERMINE NUMBER OF CANOPY HITS AND GROUND HITS PER 1 M PIXEL

Please be more specific about the calculation methods than naming the Software ArcGIS. I suppose this is an array operation which could be done in R (or any other math software) too. Which approaches did you employ? What can be understood about the "10m Buffer around the field points" (L187) and how does it differ to the "shifted square buffer" (L188f.)?

THESE CALCULATIONS COULD BE PERFORMED IN R OR ANY OTHER SOFTWARE BUT IT IS QUITE SIMPLE TO DO IN ARCGIS. THE SPECIFIC OPERATIONS INCLUDED USING THE BUFFER TOOL AND SUMMING THE NUMBER OF CANOPY AND GROUND POINTS QUANTIFIED IN THE VALUE FIELD OF THE RASTER USING THE ZONAL STATISTICS TOOL. THE SHIFT CALCULATION WAS PERFORMED IN THE SAME WAY AFTER USING THE EDITOR TOOL AND MOVE COMMAND TO SHIFT THE POINTS SOUTH BY 3.42 M. THE FINAL VERSION WILL BE EDITED TO INCLUDE THIS SPECIFIC INFORMATION.

Did you average within this area for comparison?

WE SUMMED THE VALUES AS DESCRIBED ABOVE.

What are the effects on the performance of the estimates. Especially with regards to the issue of "registration errors" L277ff. would this mean that a higher resolution could be more accurate or in other words that the hemispherical photographs suffer from minor shading effects to become representative at stand scale?

YES, THIS IS VALID CONCLUSION FROM THESE RESULTS.

For a validation of the Lidar-derived solar insolation there is basically the correlation plot in Fig. 8 comparing it to pyranometer measurements. To me this does not appear very convincing to support the conclusion. By not allowing for an intercept in your linear

regression model, you define the bias-term to be zero. While this is an understandable desire in comparing two measurements which should give the same results, I do not understand your statement in L298f.

AGREED THAT THIS IS NOT WELL STATED. THE INTENTION WAS TO BE ABLE TO PREDICT INSOLATION WITH A MODEL THAT WOULD ESTIMATE INSOLATION TO BE ZERO IN AREAS WHERE NO CANOPY POINTS WERE PRESENT. THIS WILL BE CLARIFIED IN REVISION

The 16 points appear to overestimate the pyranometer references in most cases. High insolation references are underestimated. With an R2 of 0.63, I find it rather problematic to speak of accurate: L329f. "a synthetic hemispherical photograph approach accurately predict solar insolation and light transmittance".

I STRUGGLE WITH DESIGNATION A THRESHOLD FOR ACCURACY, BUT AGREE THAT THIS IS NOT VERY PRECISE TO DECLARE THIS ACCURATE WITHOUT A THRESHOLD. WILL REWORD TO SUGGEST THAT IT MAY BE ACCURATE DEPENDING ON APPLICATION.

In this respect, I moreover have difficulties to relate this back to the presented indices which leaves me with a couple of questions about the reason of their introduction in the first place. This confusion might partially stem from the mannifold usage of the term "model" in the manuscript. I would suggest to allow for a more precise terminology to differentiate regression analyses from conversion models, from indices and from spatial map models. From the title I was expecting several modelling approaches using the Lidar data, which I did not find in the manuscript.

REVIEWER 1 HAD A SIMILAR CRITICISM AND WE WILL REVISE TO INDICATE WHAT WE MEAN BY MODEL.

Coming back to the indices (Fig. 6, Tab. 3) I do not find the focus of the study specifically suitable to address these correlations.

YOUR CRITICISM IS NOT SUFFICIENTLY DETAILED FOR ME TO RESPOND IN DETAIL. I WILL SIMPLY SAY THAT THE CORRELATIONS IN MY OPINION ARE SUITABLE AS THE OBJECTIVE OF THE STUDY IS TO EVALUATE THE DIFFERENT METHODS COMPARED TO FIELD DATA.

Contrastingly, the comparison of synthetic and actual hemispherical photograph (Fig. 7) is very compelling but falls in my view a little short in its analysis and evaluation (e.g. applying this for all 16 locations). Since the validation of the "Lidar-based modelling" is rather difficult using the 16 measurements alone, maybe some further reference could be derived from remote sensing products? This could also provide the link to some of the addressed indices?

I'M NOT SURE WHAT SPECIFICALLY YOU ARE PROPOSING. WE ARE PRESENTING THIS WORK TO STAND ALONE AND CANNOT AT THIS TIME EXPAND THE SCOPE.

2 Minor comments: L28f.: why only ecological applications?

THE SCOPE OF THIS STUDY IS FOCUSED ON ECOLOGICAL APPLICATIONS.

L29: do trees really interact (so having feedbacks) with solar radiation?

I WOULD ARGUE THAT TREES INTERACT WITH PHOTONS THROUGH REFLECTION, TRANSMITTANCE, AND ABSORPTION. SHADING IS A COMBINATION OF THESE THREE EFFECTS.

C3 HESSD Interactive comment Printer-friendly version Discussion paper L36: can (solar) energy intercept with something? maybe irradiate a stream?

WILL CHANGE TO IRRADIATE

L37: how does solar irradiation limit options for forest management? I do not understand.

THE REST OF THE PARAGRAPH EXPLAINS THIS, CULMINATION IN THE FINAL

SENTENCE WHICH ANSWERS YOUR QUESTION

L48ff.: is it really necessary to describe the function of a pyranometer (at this broad level of detail)?

I DON'T THINK IT DETRACTS FROM THE PAPER. SINCE THIS IS A HYDROLOGY JOURNAL I WANT THE TECHNICAL INFORMATION TO BE WELL EXPLAINED.

L53: I do not see the difference between the time references of a direct state measurement and the photograph

IT IS IN THE DIRECT VS INDIRECT MEASUREMENT. THE DIRECT MEASUREMENT IS DEPENDENT ON THE ANGLE OF THE SUN WHILE THE INDIRECT MEASUREMENT IS NOT

L56: Depending on the type of pyranometer, diffuse radiation is directly measured too.

THIS IS REFERRING TO HEMISPHERICAL PHOTOGRAPHS

L67: Start new paragraph with "Airborne lidar..." ?

I SEE HOW IT COULD BE GOOD TO START A NEW PARAGRAPH THERE BUT THAT WOULD LEAD TO TWO VERY SHORT PARAGRAPHS AND PREFER IT AS IS.

L113f.: very confusing. please rephrase.

I'M NOT SURE WHAT IS CONFUSING. THE CITATION TO THE ORIGINAL PAPER IS ALSO THERE TO HELP READERS IF THEY ARE CONFUSED.

Fig 1: I would prefer all four Lidar models/maps instead of the grey box, which I assume to be the total Lidar dataset footprint. If you find my suggestion feasible, maybe a map of a satellite RS derived index could also be a reference here. A colourbar would be nice.

IT WOULD BE DIFFICULT TO FIT ALL FOUR MAPS IN THIS FIGURE WITHOUT MAKING THEM EXTREMELY SMALL. THE GREY FOOTPRINT AND EXPLANATION

OF THE COLORING IS IN THE CAPTION.

L200f.: What happened to the longitudinal profiles? Were they processed?

YES, THOSE ARE IN FIGURE 9.

L215: See general comment. Which exactly are THE models? do you refer to the different indices? the calculus to derive them? a model to generate the synthetic hemispherical what are the assumptions behind the comparison approach? What is the observation reference deemed as closest to the true value?

SEE COMMENT ABOVE. WILL REVISE TO MAKE THIS MORE CLEAR.

L257: model performance? in reference to what? Is a R2 to each other really a good measure?

AGREED. IT IS A POINT THAT REVIEWER 1 ALSO BROUGHT UP AND WILL BE EDITED TO REMOVE SATISFACTORY AND CLARIFY THAT R2 IS THE METHOD OF EVALUATION AND DISCUSS THE LIMITS OF THE USE OF THAT STATISTIC.

C4 HESSD Interactive comment Printer-friendly version Discussion paper L277ff.: I do not understand why this should not be desirable... actually, i find the results in fig 7 quite convincing and the sensitivity ght be quite an interesting feature. Pls. see my general comment on this, too.

IT IS UNDESIRABLE BECAUSE IT MAKES IT DIFFICULT TO EVALUATE THE AC-CURACY OF THE MODELS. THIS WILL HOPEFULLY BE MORE CLEAR WHEN THE MODEL LANGUAGE IS REVISED.
* * *

---

## Short Comment (SC3) · 14 Apr 2019

The authors present an interesting study that compares two LiDAR based techniques (i.e., a raster-based method and a synthetic hemispherical photograph approach) for estimating under canopy solar insolation, which is an important variable for predicting stream temperature dynamics. They conduct their study for sites on the heavily forested Panther Creek and its tributary located in Oregon, USA While I am generally supportive of the merits of the study the authors present, I believe they could be more precise in their language and provide more connecting details about the methods used so that their work can be replicated and advanced. I also have some specific con-

cerns about the methods in the models. Additionally, throughout the paper, there is an emphasis on the ecological implications of this work. However, stream temperature also has important implications for various biogeochemical processes. The work the authors present may be of interest to other research domains so I would recommend that the authors broaden their discussion to encompass them. I have provided some general comments and suggestions that I hope the authors will consider incorporating into their paper to address the problems I have enumerated.

General Comments 1. While the authors indicate that they used two LiDAR based approaches/models for estimating solar insolation, midway through the paper, they introduce the new term "predictors" and then switch back to models (Line 294). This is confusing. I would suggest that the authors select one term and consistently use it throughout the paper. I would actually recommend sticking to predictor since they are essentially correlating various shading surrogate indexes with measurements of solar insolation. I also think it will be good introduce the specific predictors used under each approach (i.e., raster & synthetic hemispheric photograph approaches) at the beginning of the paper so that their introduction later in the paper is not so abrupt. Under raster-based predictors they could introduce LPI, SLPI, and LAI and then introduce %Transmittance for hemispheric photograph approach. They could also discuss why they are good/suggested predictors for solar insolation citing references.

THIS IS SIMILAR TO COMMENTS MADE BY REVIEWER 1 AND 2. YOUR SPECIFIC RECOMMENDATIONS ARE WELL RECEIVED AND WILL BE INCORPORATED INTO THE REVISED MANUSCRIPT.

2. The authors conclude that the limitation of their study was the lack of more monitoring points with large insolation values and that inclusion of more of these points would have increased the model accuracy (Line 266), but the point of their study was to derive approaches for estimating solar insolation for streams with heavily forested riparian zones. This is in practice the areas where insolation estimation uncertainty is greatest. My recommendation is to make this their focus and perhaps remove the

points with higher insolation values from their regression.

AGREE WITH THE GENERAL SENTIMENT OF THIS COMMENT. THE WORDING WAS INTENDED TO INDICATE THAT IT WOULD HAVE BEEN EASY FOR US TO CHOOSE LOCATIONS WITH LOW CANOPY COVER TO INCREASE THE ACCURACY OF THE MODEL, BUT THAT WOULD HAVE NOT BEEN PARTICULARLY USEFUL. WE WILL REWORD THIS SECTION TO MAKE IT SEEM LESS LIKE A LIMITATION AND MORE A RESULT THAT SHOULD STAND ON ITS OWN. NOTE THAT THE POINTS WITH HIGHEST %TRANSMITTANCE ONLY HAD 35% SO I DON'T THINK IT'S NECESSARY TO REMOVE THOSE AS THEY AREN'T PARTICULARLY HIGH. I THINK THE ISSUE IS MORE THAT WE WERE NOT ABLE TO CAPTURE ENOUGH POINTS IN THE 15% TO 35% RANGE.

3. Throughout the paper, the authors use the word "significant" to describe differences between values conjuring up an image of statistical significance. I would recommend that the authors state the actual numerical differences or use other words.

AGREE AND THIS IS SIMILAR TO FEEDBACK GIVEN BY REVIEWER 1 AND 2.

4. While the connection between solar insolation is self-apparent. I would recommend making that connection more explicit in the paper. You could say something along the lines of "Solar radiation is a major source heat flux into streams providing up to y% of heat fluxes" and the then cite a reference.

AGREE AND WILL ADD IN SIMILAR WORDING AND A REFERENCE.

5. For the synthetic hemispherical photographs, what resolution was used for the hemisphere? Did it match the field photographs? If different, what are the implications of the differences for the authors analysis. I think the comparison of these too and the reasons why they might differ is an important contribution.

IT'S A BIT DIFFICULT TO COMPARE AS THE SYNTHETIC PHOTOGRAPHS ARE CREATED USING POINTS THAT ARE RENDERED WITH A RELATIVELY LARGE

"DOT" SIZE COMPARED TO THE INDIVIDUAL PIXELS OF THE CAMERA. THE "DOT" SIZE WAS DETERMINED BY THE MOESER ET AL (2014) ALGORITHM. THE INTENTION IS FOR THE READER TO USE FIGURE 7 TO JUDGE THESE DIFFERENCES. I AM NOT SURE HOW DIFFERENCES IN RESOLUTION WOULD AFFECT THE ANALYSIS.

Specific Comments 1. Line 16 – "due to the importance of temperature to aquatic biota". This makes it sound like aquatic biota is the only reason why quantifying solar insolation is important. Consider revising to broaden its implications.

IT WAS OUR MAIN MOTIVATION FOR EMBARKING ON THIS STUDY, BUT IT DOES LIMIT ITS IMPLICATIONS. WILL CHANGE TO "USEFUL FOR A VARIETY OF APPLICATIONS, AND A SPECIFIC FOCUS OF THIS STUDY IS THE IMPORTANCE OF STREAM TEMPERATURE TO AQUATIC BIOTA.

2. Line 17-19: I suggest changing "two approaches. . ." to something like "four predictor indexes computed using two approaches for estimate shading effects from LiDAR" or something along these lines. The larger point is that it is important to be precise in describing what was actually done.

AGREED. WE WILL MAKE THIS CHANGE.

3. Line 28 "is essential to a diversity of ecological. . ." Again, I think you can broaden this.

WILL ADD ANOTHER SENTENCE TO BROADEN THE SCOPE BEFORE FOCUSING ON ECOLOGICAL APPLICATIONS.

4. Line 36 "solar energy intercepting a stream. . ." Consider revising to "solar energy irradiating a stream"

SAME COMMENT WAS MADE BY REVIEWER 2 AND IT WILL BE CHANGED.

5. Line 36-37 "can in turn limit options for forest management". Could the authors

explain how increasing temperatures limit options for forest management? I am not sure this is true.

A SIMILAR COMMENT WAS MADE BY REVIEWER 2. UPON FURTHER REFLECTION WE SEE HOW THIS SENTENCE IS CONFUSING AND WILL EDIT IT TO MAKE THE CONNECTION BETWEEN STREAM TEMPERATURES AND THE REQUIREMENT TO KEEP UNHARVESTABLE BUFFERS NEAR STREAMS

6. Line 45-46 "models may be needed..." I would argue that this is actually often the approach that is used and is not a new insight so please consider revising to "models are therefore often employed to estimate temperature"

GOOD POINT. WILL CHANGE TO ADOPT THAT LANGUAGE

7. Line 57: "solar output" consider revising to extra-terrestrial solar radiation.

WILL CHANGE. THANKS!

8. Line 60: "All ground-based. . ." Sounds a little too strong. Consider removing "All".

AGREED. WILL CHANGE

9. Line 78-79. "GIS software solar radiation calculators. . ." Consider revising to "Solar radiation calculators in GIS software"

GOOD EDIT. WILL CHANGE.

10. Line 80-82. I think you are missing some words somewhere. Please rephrase for clarity. E.g., "r.sun solar insolation model for the GRASS GIS software. . ."

AGREED THAT THIS IS PHRASED POORLY. WILL REWORD.

11. Line 89: What are Ellenburg indicator values? While ecologist might be familiar with them, I think it will be good to explain.

WILL ADD A SHORT DESCRIPTION.

12. Line 169 Figure 4: Does the y axis name need to be solar irradiance for consistency?

GOOD CATCH. WILL CHANGE.

13. Line 195-197: I am not sure why this sentence is part of the paper. I feel it is unnecessary. Please consider removing.

THE METHOD THAT WE USED BASED ON BODE ET AL (2014) USED THIS TOPOGRAPHIC CORRECTION AND WE WANTED TO EXPLAIN WHY WE DID NOT FOLLOW THEIR METHOD COMPLETELY.

14. Line 198-199: Are the authors able to delve more into the details of the creation of these synthetic photos?

THE CODE USED TO CREATE THESE WAS SHARED WITH PERMISSION BY DAVE MOESER AND WOULD REFER YOU TO HIM FOR FURTHER DETAIL.

15. Line 222. "significantly improved" remove significantly for the reasons I raised earlier.

AGREED

16. Line 278: Please remove the word "significant". for the same reasons as before.

AGREED

17. Line 298-299: I am not sure I am comfortable removing the intercept and saying the resulting model has little bias. By removing the intercept, the authors are making the RËȨ2 value no longer useful.

THE INTERCEPT WAS REMOVED SO THAT PIXELS WITH NO CANOPY POINTS WOULD YIELD A PREDICTED VALUE OF 0. WILL MAKE THIS REASONING EXPLICIT IN THE REVISED VERSION.

18. Line 311 & Figure 9: Please consider adding an inset that zooms to one of the

monitoring points.

I AM NOT SURE WHAT YOU MEAN BY MONITORING POINTS. ARE YOU SUGGESTING AN INSET SIMILAR TO FIGURE 1? IF SO, I DON'T THINK A SIMILAR INSET WOULD BE PARTICULARLY USEFUL FOR INTERPRETATION OF FIGURE 9.

19. Line 337-340: The authors pivots to ray tracing. However, the methods they use does not include any ray tracing.

THIS POINT WAS BROUGHT UP BY REVIEWER 1 AND WE AGREE THAT IT DOES NOT BELONG. IT WILL BE EDITED TO INCLUDE ONLY A SHORT REFERENCE TO RAY TRACING AS A POTENTIAL AVENUE OF FUTURE RESEARCH

---

## Short Comment (SC4) · 14 Apr 2019

Please note that the reviewer's comment about Line 37 has been re-evaluated based on additional comments by Reviewer 3 and line 37 will be edited to make the connection more explicit.

---

## Author Comment (AC1) · 30 Apr 2019

All co-authors agree with the response written in SC1.
* * *

---

## Author Comment (AC2) · 30 Apr 2019

All co-authors agree with the response written in SC2 and SC4.

---

## Author Comment (AC3) · 30 Apr 2019

All co-authors agree with the response written in SC3.
* * *

---

## Author Response (AR1)

**Response to Editor**

On top of the comments made by the referees, I also have a suggestion in combination with Fig. 7 which shows the synthetic and a field acquired hemispherical photograph. Since this figure shows qualitatively the sensitivity to small errors in the exact location, it would be beneficial to show the daily timeserie of the pyranometer (as in Fig. 4) with the simulated timeseries of a couple of points surrounding the pyranometer. This may show that points located 1 or 2 meters away from the pyranometer give simulated solar radiation similar to the observed one.

WE HAVE ADDED A FIGURE 8 WHICH SHOWS THE PYRANOMETER TIMESERIES FOR FIGURE 7. THE PREDICTED SOLAR RADIATION IS ACTUALLY VERY SIMILAR TO THE OBSERVED PYRANOMTER DATA (0.58 VS 0.55 VS 0.57 kW/m$^2$.  IT WAS CHOSEN BECAUSE IT WAS PARTICULARY EASY TO OBSERVE HOW SMALL LOCATION ERRORS COULD POTENTIALLY AFFECT THE SIMULATED SUNPATH. WE DO NOT HAVE A METHODOLOGY TO SIMULATE A TIMESERIES, BUT WE ARE WILLING TO DO ADDITIONAL ANALYSES IF YOU THINK THEY WOULD BE HELPFUL.

In stream temperature models it often does not matter that much if the exact location of solar insolation is shifted a few meters.

Having said this, I was also wondering if you have taken into account that the pyranometer was located 1 m above the surface. Especially if solar angels are low, this may influence your result.

THE HEMISPHERICAL PHOTOGRAPH AND PYRANOMETER WERE TAKEN AT THE SAME HEIGHT, BUT IT IS A GOOD POINT THAT THE ADDITIONAL METER IS NOT PROPERLY ACCOUNTED FOR IN COMPARING LIDRA AND THE FIELD METHODS. WE WILL ADD LANGAUAGE POINTING OUT THIS COULD BE A SOURCE OF ERROR.

**Response to Reviewer 1**

On three substantive issues I have concerns: Model vs predictor. The abstract clearly states this paper is testing two models with two validation datasets. However, under Model Comparisons, the discussion changes to four "predictors" without explanation how these relate to the two models or why effective leaf area index is included, as it is part of neither model. This confusion is compounded under Model Application, where the predictors are now referred to as Model G and Model E, in reference to graphs in figure 6. More consistent naming from methods through the discussion would make this easier to follow.

AGREED THAT THIS IS CONFUSING AND IMPRECISE. THE FINAL VERSION WILL BE EDITED TO CLARIFY THE EXACT PREDICTORS USED IN THE ABSTRACT, METHODS, RESULTS AND DISCUSSION.

Pyranometer validation. The spectral response of silicon-cell photodiodes is calibrated to clear sky
direct sunlight conditions, because it is not sensitive to the full shortwave spectrum and responds to
various wavelengths with different intensities. Leaf shading selectively blocks certain wavelengths,
which causes silicon pyranometers to decalibrate. Apogee estimates that this produces roughly a 19%
error under conifer canopy (https://www.apogeeinstruments.com/content/SP-100-200- specsheet.pdf,
page 15). Black body thermopile pyranometers are recommended for subcanopy light measurements.
They have an even spectral response across the shortwave spectrum even under leaves. I recommend
the authors acknowledge this as a source of uncertainty in their discussion.

THANK YOU FOR POINTING THIS OUT. WE WILL ADD THIS SOURCE OF UNCERTAINTY TO THE
DISCUSSION.

Conclusions. Line 256 "While both the raster-based LPI approach and the lidar point reprojection
synthetic hemispherical photograph approach achieve satisfactory model performance, the limited
range of solar insolation conditions at the point locations in our study limits some of the conclusions
that an be drawn." While I appreciate this study and the intent behind it, perhaps more validation data
is needed? Was there insufficient information to effectively evaluate the two models? How are both
approaches satisfactory

AGREED THAT "SATISFACTORY" IS NOT WELL-DEFINED AND THUS THIS STATEMENT IS NOT VERY
USEFUL. WILL REWORD TO INDICATE THAT THE RESULTS MAY BE SATISFACTORY DEPENDING ON THE
APPLICATION BUT MORE VALIDATION DATA IS NEEDED.

SPECIFIC COMMENTS

Line 146: The dates are not given for when the pyranometers were recorded. This makes a significant
difference for the models. On June 20, summer solstice, the shifted LPI and general LPI will look almost
identical, but December 20, winter solstice, will look radically different. Is there a reason this is not
mentioned, while the date for the Lidar is mentioned?

THIS WAS AN OVERSIGHT. PYRANOMETER AND HEMIPHOTO DATA WERE COLLECTED OVER TWO WEEKS
AROUND THE SUMMER SOLSTICE IN 2015. THIS INFORMATION WILL BE ADDED TO THE METHODS.

Line 251: Table 3 linear regression slope and intercept. I think this can be removed without loss to the
paper.

THIS IS INCLUDED FOR COMPLETENESS SAKE AND BECAUSE CERTAIN SCATTER PLOTS IN FIGURE 6 (eg. G
AND H MIGHT BE DIFFICULT TO INTERPRET WITHOUT THE INCLUSION OF A 1:1 LINE)

Line 269: Models should agree better in areas without shading. I am not sure how this is a conclusion.
While true, the whole point of these models is to tackle the uncertainty of heavily shaded landscapes.

THAT SENTENCE WILL BE REMOVED

Line 271: small registration errors. Recommend identifying which model this is an error for. Relevant for
synthetic photo, but not for raster.

AGREED. WILL INCLUDE IN REVISED VERSION

Line 281: understory vegetation. This is actually an argument against the directions this paper
recommends on Line 335 regarding ray tracing. Note the raster approach was developed with this issue
as one of the problems it was solving in its design.

GOOD OBSERVATION AND AGREED. WILL REMOVE RAY TRACING FROM THE CONCLUSION EXCEPT TO
NOTE THAT FURTHER RESEARCH IS NEEDED.

Line 294: "Model G and Model E (figure 6) performed the best..." This statement is unclear. How are
plots models? What criteria states that they performed the best? Their performance and the
performance of the hemispherical photos all seem within error of each other. Is this incorrect?

THIS CONCLUSION WAS BASED ON THE COEFFICIENT OF DETERMINATION. WE ARE CONSIDERING THE
SIMPLE LINEAR REGRESSION AS THE MODEL.

Line 337: "The results of this study suggest that refined ray-tracing approaches should not require
calibration." I do not see this statement supported by the paper. Both models used in this study did not
perform point cloud ray tracing. That is their strength. Musselman and Lee (referenced in introduction)
used voxel ray-tracing. Both required calibration.

AGREED. THIS SENTENCE WILL BE REMOVED AND RAY TRACING WILL BE REMOVED FROM CONCLUSION
EXCEPT FOR SHORT STATEMENT ON FURTHER RESEARCH.

**RESPONSE TO REVIEWER 2**

I appreciate very much that the authors provide their data and analysis (as is HESS standard now). While
I could easily follow the general setup of the study, I found it difficult to grasp the information residing in
the Lidar data set and how it has been used. Since the latter is not included in the repository: Did I
understand correctly that the Lidar data was commercially acquired and preprocessed to 1m pixels? So
each pixel has values about all point returns, the number of highest hits (canopy) and the number of
lowest hits (ground)?

THE LIDAR WAS PRE-PROCESSED BY THE VENDOR INTO 1 M PIXELS CONTAINING HIGHEST ELEVATION
AND GROUND MODEL. THE AUTHORS CREATED ADDITIONAL RASTERS USING THE RAW LIDAR POINTS IN
ORDER TO DETERMINE NUMBER OF CANOPY HITS AND GROUND HITS PER 1 M PIXEL

Please be more specific about the calculation methods than naming the Software ArcGIS. I suppose this
is an array operation which could be done in R (or any other math software) too. Which approaches did you employ? What can be understood about the "10m Buffer around the field points" (L187) and how
does it differ to the "shifted square buffer" (L188f.)?

THESE CALCULATIONS COULD BE PERFORMED IN R OR ANY OTHER SOFTWARE BUT IT IS QUITE SIMPLE
TO DO IN ARCGIS. THE SPECIFIC OPERATIONS INCLUDED USING THE BUFFER TOOL AND SUMMING THE
NUMBER OF CANOPY AND GROUND POINTS QUANTIFIED IN THE VALUE FIELD OF THE RASTER USING
THE ZONAL STATISTICS TOOL. THE SHIFT CALCULATION WAS PERFORMED IN THE SAME WAY AFTER
USING THE EDITOR TOOL AND MOVE COMMAND TO SHIFT THE POINTS SOUTH BY 3.42 M. THE FINAL
VERSION WILL BE EDITED TO INCLUDE THIS SPECIFIC INFORMATION.

Did you average within this area for comparison?

WE SUMMED THE VALUES AS DESCRIBED ABOVE.

What are the effects on the performance of the estimates. Especially with regards to the issue of
"registration errors" L277ff. would this mean that a higher resolution could be more accurate or in other
words that the hemispherical photographs suffer from minor shading effects to become representative
at stand scale?

YES, THIS IS VALID CONCLUSION FROM THESE RESULTS.

For a validation of the Lidar-derived solar insolation there is basically the correlation plot in Fig. 8
comparing it to pyranometer measurements. To me this does not appear very convincing to support the
conclusion. By not allowing for an intercept in your linear regression model, you define the bias-term to
be zero. While this is an understandable desire in comparing two measurements which should give the
same results, I do not understand your statement in L298f.

AGREED THAT THIS IS NOT WELL STATED. THE INTENTION WAS TO BE ABLE TO PREDICT INSOLATION
WITH A MODEL THAT WOULD ESTIMATE INSOLATION TO BE ZERO IN AREAS WHERE NO CANOPY POINTS
WERE PRESENT. THIS WILL BE CLARIFIED IN REVISION

The 16 points appear to overestimate the pyranometer references in most cases. High insolation
references are underestimated. With an R2 of 0.63, I find it rather problematic to speak of accurate:
L329f. "a synthetic hemispherical photograph approach accurately predict solar insolation and light
transmittance".

WE STRUGGLE WITH DESIGNATING A THRESHOLD FOR ACCURACY, BUT AGREE THAT THIS IS NOT VERY
PRECISE TO DECLARE THIS ACCURATE WITHOUT A THRESHOLD. WILL REWORD TO SUGGEST THAT IT
MAY BE ACCURATE DEPENDING ON APPLICATION.

In this respect, I moreover have difficulties to relate this back to the presented indices which leaves me
with a couple of questions about the reason of their introduction in the first place. This confusion might
partially stem from the mannifold usage of the term "model" in the manuscript. I would suggest to allow
for a more precise terminology to differentiate regression analyses from conversion models, from
indices and from spatial map models. From the title I was expecting several modelling approaches using
the Lidar data, which I did not find in the manuscript.

REVIEWER 1 HAD A SIMILAR CRITICISM AND WE WILL REVISE TO INDICATE WHAT WE MEAN BY MODEL.

Coming back to the indices (Fig. 6, Tab. 3) I do not find the focus of the study specifically suitable to
address these correlations.

YOUR CRITICISM IS NOT SUFFICIENTLY DETAILED FOR ME TO RESPOND IN DETAIL. I WILL SIMPLY SAY
THAT THE CORRELATIONS IN MY OPINION ARE SUITABLE AS THE OBJECTIVE OF THE STUDY IS TO
EVALUATE THE DIFFERENT METHODS COMPARED TO FIELD DATA.

Contrastingly, the comparison of synthetic and actual hemispherical photograph (Fig. 7) is very
compelling but falls in my view a little short in its analysis and evaluation (e.g. applying this for all 16
locations). Since the validation of the "Lidar-based modelling" is rather difficult using the 16
measurements alone, maybe some further reference could be derived from remote sensing products?
This could also provide the link to some of the addressed indices?

I'M NOT SURE WHAT SPECIFICALLY YOU ARE PROPOSING. WE ARE PRESENTING THIS WORK TO STAND
ALONE AND CANNOT AT THIS TIME EXPAND THE SCOPE.

2 Minor comments: L28f.: why only ecological applications?

THE SCOPE OF THIS STUDY IS FOCUSED ON ECOLOGICAL APPLICATIONS.

L29: do trees really interact (so having feedbacks) with solar radiation?

I WOULD ARGUE THAT TREES INTERACT WITH PHOTONS THROUGH REFLECTION, TRANSMITTANCE, AND
ABSORPTION. SHADING IS A COMBINATION OF THESE THREE EFFECTS.

L36: can (solar) energy intercept with something? maybe irradiate a stream?

WILL CHANGE TO IRRADIATE

L37: how does solar irradiation limit options for forest management? I do not understand.

THE REST OF THE PARAGRAPH EXPLAINS THIS, CULMINATING IN THE FINAL SENTENCE WHICH ANSWERS
YOUR QUESTION

L48ff.: is it really necessary to describe the function of a pyranometer (at this broad level of detail)?

I DON'T THINK IT DETRACTS FROM THE PAPER. SINCE THIS IS A HYDROLOGY JOURNAL I WANT THE
TECHNICAL INFORMATION TO BE WELL EXPLAINED.

L53: I do not see the difference between the time references of a direct state measurement and the
photograph

IT IS IN THE DIRECT VS INDIRECT MEASUREMENT. THE DIRECT MEASUREMENT IS DEPENDENT ON THE
ANGLE OF THE SUN WHILE THE INDIRECT MEASUREMENT IS NOT

L56: Depending on the type of pyranometer, diffuse radiation is directly measured too.

THIS IS REFERRING TO HEMISPHERICAL PHOTOGRAPHS

L67: Start new paragraph with "Airborne lidar..." ?

I SEE HOW IT COULD BE GOOD TO START A NEW PARAGRAPH THERE BUT THAT WOULD LEAD TO TWO
VERY SHORT PARAGRAPHS AND PREFER IT AS IS.

L113f.: very confusing. please rephrase.

I'M NOT SURE WHAT IS CONFUSING. THE CITATION TO THE ORIGINAL PAPER IS ALSO THERE TO HELP
READERS IF THEY ARE CONFUSED.

Fig 1: I would prefer all four Lidar models/maps instead of the grey box, which I assume to be the total
Lidar dataset footprint. If you find my suggestion feasible, maybe a map of a satellite RS derived index
could also be a reference here. A colourbar would be nice.

IT WOULD BE DIFFICULT TO FIT ALL FOUR MAPS IN THIS FIGURE WITHOUT MAKING THEM EXTREMELY
SMALL. THE GREY FOOTPRINT AND EXPLANATION OF THE COLORING IS IN THE CAPTION.

L200f.: What happened to the longitudinal profiles? Were they processed?

YES, THOSE ARE IN FIGURE 10.

L215: See general comment. Which exactly are THE models? do you refer to the different indices? the
calculus to derive them? a model to generate the synthetic hemispherical what are the assumptions
behind the comparison approach? What is the observation reference deemed as closest to the true
value?

SEE COMMENT ABOVE. WILL REVISE TO MAKE THIS MORE CLEAR.

L257: model performance? in reference to what? Is a R2 to each other really a good measure?

AGREED. IT IS A POINT THAT REVIEWER 1 ALSO BROUGHT UP AND WILL BE EDITED TO REMOVE
SATISFACTORY AND CLARIFY THAT R2 IS THE METHOD OF EVALUATION AND DISCUSS THE LIMITS OF THE
USE OF THAT STATISTIC.

L277ff.: I do not understand why this should not be desirable... actually, i find the results in fig 7 quite
convincing and the sensitivity ght be quite an interesting feature. Pls. see my general comment on this,
too.

IT IS UNDESIRABLE BECAUSE IT MAKES IT DIFFICULT TO EVALUATE THE ACCURACY OF THE MODELS. THIS
WILL HOPEFULLY BE MORE CLEAR WHEN THE MODEL LANGUAGE IS REVISED.

**RESPONSE TO REVIEWER 3**

The authors present an interesting study that compares two LiDAR based techniques (i.e., a raster-based
method and a synthetic hemispherical photograph approach) for estimating under canopy solar
insolation, which is an important variable for predicting stream temperature dynamics. They conduct
their study for sites on the heavily forested Panther Creek and its tributary located in Oregon, USA While
I am generally supportive of the merits of the study the authors present, I believe they could be more
precise in their language and provide more connecting details about the methods used so that their
work can be replicated and advanced. I also have some specific concerns about the methods in the
models. Additionally, throughout the paper, there is an emphasis on the ecological implications of this
work. However, stream temperature also has important implications for various biogeochemical
processes. The work the authors present may be of interest to other research domains so I would
recommend that the authors broaden their discussion to encompass them. I have provided some
general comments and suggestions that I hope the authors will consider incorporating into their paper
to address the problems I have enumerated.

General Comments 1. While the authors indicate that they used two LiDAR based approaches/models
for estimating solar insolation, midway through the paper, they introduce the new term "predictors"
and then switch back to models (Line 294). This is confusing. I would suggest that the authors select one
term and consistently use it throughout the paper. I would actually recommend sticking to predictor
since they are essentially correlating various shading surrogate indexes with measurements of solar
insolation. I also think it will be good introduce the specific predictors used under each approach (i.e.,
raster & synthetic hemispheric photograph approaches) at the beginning of the paper so that their
introduction later in the paper is not so abrupt. Under raster-based predictors they could introduce LPI,
SLPI, and LAI and then introduce %Transmittance for hemispheric photograph approach. They could also
discuss why they are good/suggested predictors for solar insolation citing references.

THIS IS SIMILAR TO COMMENTS MADE BY REVIEWER 1 AND 2. YOUR SPECIFIC RECOMMENDATIONS ARE
WELL RECEIVED AND WILL BE INCORPORATED INTO THE REVISED MANUSCRIPT.

2. The authors conclude that the limitation of their study was the lack of more monitoring points with
large insolation values and that inclusion of more of these points would have increased the model
accuracy (Line 266), but the point of their study was to derive approaches for estimating solar insolation
for streams with heavily forested riparian zones. This is in practice the areas where insolation estimation
uncertainty is greatest. My recommendation is to make this their focus and perhaps remove the points
with higher insolation values from their regression.

AGREE WITH THE GENERAL SENTIMENT OF THIS COMMENT. THE WORDING WAS INTENDED TO
INDICATE THAT IT WOULD HAVE BEEN EASY FOR US TO CHOOSE LOCATIONS WITH LOW CANOPY COVER

TO INCREASE THE ACCURACY OF THE MODEL, BUT THAT WOULD HAVE NOT BEEN PARTICULARLY
USEFUL. WE WILL REWORD THIS SECTION TO MAKE IT SEEM LESS LIKE A LIMITATION AND MORE A
RESULT THAT SHOULD STAND ON ITS OWN. NOTE THAT THE POINTS WITH HIGHEST %TRANSMITTANCE
ONLY HAD 35% SO WE DON'T THINK IT'S NECESSARY TO REMOVE THOSE AS THEY AREN'T
PARTICULARLY HIGH. WETHINK THE ISSUE IS MORE THAT WE WERE NOT ABLE TO CAPTURE ENOUGH
POINTS IN THE 15% TO 35% RANGE.

3. Throughout the paper, the authors use the word "significant" to describe differences between values
conjuring up an image of statistical significance. I would recommend that the authors state the actual
numerical differences or use other words.

AGREE AND THIS IS SIMILAR TO FEEDBACK GIVEN BY REVIEWER 1 AND 2.

4.

While the connection between solar insolation is self-apparent. I would recommend making that
connection more explicit in the paper. You could say something along the lines of "Solar radiation is a
major source heat flux into streams providing up to y% of heat fluxes" and the then cite a reference.

AGREE AND WILL ADD IN SIMILAR WORDING.

5. For the synthetic hemispherical photographs, what resolution was used for the hemisphere? Did it
match the field photographs? If different, what are the implications of the differences for the authors
analysis. I think the comparison of these too and the reasons why they might differ is an important
contribution.

IT'S A BIT DIFFICULT TO COMPARE AS THE SYNTHETIC PHOTOGRAPHS ARE CREATED USING POINTS THAT
ARE RENDERED WITH A RELATIVELY LARGE "DOT" SIZE COMPARED TO THE INDIVIDUAL PIXELS OF THE
CAMERA. THE "DOT" SIZE WAS DETERMINED BY THE MOESER ET AL (2014) ALGORITHM. THE INTENTION
IS FOR THE READER TO USE FIGURE 7 TO JUDGE THESE DIFFERENCES. WE ARE NOT SURE HOW
DIFFERENCES IN RESOLUTION WOULD AFFECT THE ANALYSIS.

Specific Comments 1. Line 16 – "due to the importance of temperature to aquatic biota". This makes it
sound like aquatic biota is the only reason why quantifying solar insolation is important. Consider
revising to broaden its implications.

IT WAS OUR MAIN MOTIVATION FOR EMBARKING ON THIS STUDY, BUT IT DOES LIMIT ITS
IMPLICATIONS. WILL CHANGE TO "USEFUL FOR A VARIETY OF APPLICATIONS, AND A SPECIFIC FOCUS OF
THIS STUDY IS THE IMPORTANCE OF STREAM TEMPERATURE TO AQUATIC BIOTA.

2. Line 17-19: I suggest changing "two approaches. . ." to something like "four predictor indexes
computed using two approaches for estimate shading effects from LiDAR" or something along these
lines. The larger point is that it is important to be precise in describing what was actually done.

AGREED. WE WILL MAKE THIS CHANGE.

3. Line 28 "is essential to a diversity of ecological. . ." Again, I think you can broaden this.

WILL ADD ANOTHER SENTENCE TO BROADEN THE SCOPE BEFORE FOCUSING ON ECOLOGICAL
APPLICATIONS.

4. Line 36 "solar energy intercepting a stream. . ." Consider revising to "solar energy irradiating a
stream"

SAME COMMENT WAS MADE BY REVIEWER 2 AND IT WILL BE CHANGED. 5.

Line 36-37 "can in turn limit options for forest management". Could the authors explain how increasing
temperatures limit options for forest management? I am not sure this is true.

A SIMILAR COMMENT WAS MADE BY REVIEWER 2. UPON FURTHER REFLECTION WE SEE HOW THIS
SENTENCE IS CONFUSING AND WILL EDIT IT TO MAKE THE CONNECTION BETWEEN STREAM
TEMPERATURES AND THE REQUIREMENT TO KEEP UNHARVESTABLE BUFFERS NEAR STREAMS

6. Line 45-46 "models may be needed..." I would argue that this is actually often the approach that is
used and is not a new insight so please consider revising to "models are therefore often employed to
estimate temperature"

GOOD POINT. WILL CHANGE TO ADOPT THAT LANGUAGE

7. Line 57: "solar output" consider revising to extra-terrestrial solar radiation.

WILL CHANGE. THANKS!

8. Line 60: "All ground-based. . ." Sounds a little too strong. Consider removing "All".

AGREED. WILL CHANGE

9. Line 78-79. "GIS software solar radiation calculators. . ." Consider revising to "Solar radiation
calculators in GIS software"

GOOD EDIT. WILL CHANGE.

10. Line 80-82. I think you are missing some words somewhere. Please rephrase for clarity. E.g., "r.sun
solar insolation model for the GRASS GIS software. . ."

AGREED THAT THIS IS PHRASED POORLY. WILL REWORD.

11. Line 89: What are Ellenburg indicator values? While ecologist might be familiar with them, I think it
will be good to explain.

WILL ADD A SHORT DESCRIPTION.

12. Line 169 Figure 4: Does the y axis name need to be solar irradiance for consistency? GOOD CATCH.
WILL CHANGE.

13. Line 195-197: I am not sure why this sentence is part of the paper. I feel it is unnecessary. Please
consider removing.

THE METHOD THAT WE USED BASED ON BODE ET AL (2014) USED THIS TOPOGRAPHIC CORRECTION
AND WE WANTED TO EXPLAIN WHY WE DID NOT FOLLOW THEIR METHOD COMPLETELY.

14. Line 198-199: Are the authors able to delve more into the details of the creation of these synthetic
photos?

THE CODE USED TO CREATE THESE WAS SHARED WITH PERMISSION BY DAVE MOESER AND WOULD
REFER YOU TO HIM FOR FURTHER DETAIL.

15. Line 222. "significantly improved" remove significantly for the reasons I raised earlier.

AGREED

16. Line 278: Please remove the word "significant". for the same reasons as before.

AGREED

17. Line 298-299: I am not sure I am comfortable removing the intercept and saying the resulting model
has little bias. By removing the intercept, the authors are making the RË ¸E2 value no longer useful.

THE INTERCEPT WAS REMOVED SO THAT PIXELS WITH NO CANOPY POINTS WOULD YIELD A PREDICTED
VALUE OF 0. WILL MAKE THIS REASONING EXPLICIT IN THE REVISED VERSION.

18. Line 311 & Figure 9: Please consider adding an inset that zooms to one of the monitoring points.

WE ARE NOT SURE WHAT YOU MEAN BY MONITORING POINTS. ARE YOU SUGGESTING AN INSET
SIMILAR TO FIGURE 1? IF SO, WE DON'T THINK A SIMILAR INSET WOULD BE PARTICULARLY USEFUL FOR
INTERPRETATION OF FIGURE

9. 19. Line 337-340: The authors pivots to ray tracing. However, the methods they use does not include
any ray tracing.

THIS POINT WAS BROUGHT UP BY REVIEWER 1 AND WE AGREE THAT IT DOES NOT BELONG. IT WILL BE
EDITED TO INCLUDE ONLY A SHORT REFERENCE TO RAY TRACING AS A POTENTIAL AVENUE OF FUTURE
RESEARCH

Lidar-based  approaches for estimating solar insolation in heavily forested streams

Richardson, Jeffrey J.*[1]; Torgersen, Christian E.[2]; and Moskal, L. Monika[3]

[1] *Sterling College, Craftsbury Common, VT, USA*

[2]*U.S. Geological Survey, Forest and Rangeland Ecosystem Science Center, Cascadia Field Station, University*

*of Washington, Seattle, WA, USA*

[3] *Precision Forestry Cooperative, School of Environmental and Forest Science, University of Washington,*

*Seattle, WA, USA*

*Corresponding Author

*This draft manuscript is distributed solely for the purposes of scientific peer review. Its content is*

*deliberative and predecisional, so it must not be disclosed or released by reviewers. Because the*

*manuscript has not yet been approved for publication by the U.S. Geological Survey (USGS), it does not*

*represent any official USGS finding or policy.*

[revised manuscript text omitted]